# tensorFM: Low-Rank Approximations of Cross-Order Feature Interactions

## Abstract

We address prediction problems on tabular categorical data, where each instance is defined by multiple categorical attributes, each taking values from a finite set. These attributes are often referred to as fields, and their categorical values as features. Such problems frequently arise in practical applications, including click-through rate prediction and social sciences. We introduce and analyze tensorFM, a new model that efficiently captures high-order interactions between attributes via a low-rank tensor approximation representing the strength of these interactions. Our model generalizes field-weighted factorization machines. Empirically, tensorFM demonstrates competitive performance with state-of-the-art methods. Additionally, its low latency makes it well-suited for time-sensitive applications, such as online advertising.

## 1 Introduction

Learning and knowledge discovery from tabular and categorical data is an important problem with various application areas. We assume that each instance is characterized by multiple categorical attributes, each with a finite domain. These attributes are often referred to as fields, and their categorical values as features. Even when the original attribute domains are numeric, they can be discretized into nominal values. This approach not only enhances model interpretability but also enables the modeling of more complex relationships.

Consider the following two notable and diverse examples of applications where tabular categorical data are highly common. In social sciences, demographic data and categorical properties often describe instances. For example, when studying recidivism, typical attributes include age, race, gender, charge degree, number of prior criminal records, and re-arrest indicator. While some of these attributes, such as age or prior criminal records, are numeric, discretization can be beneficial, as discussed earlier. In such studies, it is crucial to develop interpretable models whose decision-making processes can be easily understood, and can reveal interdependencies between attributes. In online advertising, when predicting the click-through rate, useful attributes include user ID, geolocation, time of day, advertiser ID, and top-level domain. Similar to the previous example, these attributes are either categorical or can benefit from discretization. However, a key challenge in this domain is the limited availability of memory and computational power during real-time serving. Since the model must process a vast number of ad requests per second, this constraint necessitates a trade-off between accuracy and model size. Table 1 gives an example of data for both discussed problems.

Table 1: Examples of multi-field categorical data: recidivism (left) and online advertising (right).

| Recidivism | Age | Gender | Charge | Re-arrest | | Click | User ID | Country | Hour | Adv. ID |
|---|---|---|---|---|---|---|---|---|---|---|
| -1 | 24 | M | Felony | 0 | | 1 | 1404801 | Italy | 14 | 182837 |
| -1 | 34 | F | Felony | 0 | | -1 | 4013723 | France | 19 | 172726 |
| 1 | 19 | M | Misd. | 1 | | -1 | 5200686 | France | 5 | 182823 |
| 1 | 25 | F | Felony | 0 | | 1 | 2190445 | Germany | 12 | 937362 |

Problems of this type can be effectively addressed using Factorization Machines (FMs), as they can model pairwise interactions between attributes, generate feature embeddings, and maintain computational efficiency.

In fact, FMs have gained significant attention in online advertising and recommender systems due to their high performance and efficiency. Since their introduction, various FM variants have been proposed (e.g. Juan et al., 2017; Pan et al., 2018; Wang et al., 2017), all aiming to explicitly model second-order feature interactions.

The key element of FMs is the low-rank approximation of the weight matrix that captures second-order feature interactions. Specifically, each feature is associated with a low-dimensional embedding vector, and the interaction between two features is computed as the dot product of their respective embedding vectors. A limitation of this approach is that a feature's embedding is used to model the interactions with the embedding of the features from all the other fields. However, in many applications, the strength of interactions between the same features can vary across different fields. For example, the interaction between a top-level domain and gender may be strong, whereas gender might interact only weakly with country.

Field-aware Factorization Machines (FFM) (Juan et al., 2017) address this issue by assigning distinct embeddings for each feature-field interaction. This strategy allows for a more nuanced modeling of the interactions, but significantly increases the complexity of the model. To mitigate this, Field-weighted Factorization Machine FwFM (Pan et al., 2018) uses instead an additional *interaction strength matrix* that weights interactions between fields. While this approach reduces the number of parameters compared to FFMs, it still requires iterating over all possible pairs of fields, which can be computationally expensive for large inputs. Adding structure to the interaction strength matrix, such as a low-rank constraint, reduces the computational cost of evaluating pairwise field interactions Almagor & Hoshen (2022); Shtoff et al. (2024b).

The power of deep neural networks to implicitly learn feature interactions has also been considered (Cheng et al., 2016; Guo et al., 2017). A further interesting research direction studies two-stream approaches that combine a deep neural network with models that explicitly compute feature interactions (Lian et al., 2018; Wang et al., 2021; Mao et al., 2023; Coleman et al., 2024). While these approaches can yield state-of-the-art performance, the inherent complexity of these deep architectures results in larger inference times.

Given the constraints discussed above, we focus on models that learn explicit feature interactions without a deep neural network component. Our goal is to design a model that has a small number of parameters and exhibits a small inference time. Our work builds on the Field-weighted Factorization Machine (FwFM) model, which learns a matrix to weight interactions between different fields. Following prior work, we first impose a low-rank structure on this matrix, enabling efficient evaluation of second-order feature interactions. Our main contribution is to extend this approach to higher-order interactions by applying a Canonical Polyadic (CP) decomposition to a tensor encoding the interaction strengths. Based on this idea, we introduce a variant of the FwFM model that allows the evaluation of higher-order interactions while retaining the feasibility of fast online inference. Our experimental findings indicate that the proposed method achieves a favorable trade-off between accuracy and prediction time compared to existing approaches in the literature.

Our contributions can be summarized as follows:

1. We introduce a new model dubbed tensorFM, which is a higher-order variant of FwFM with a low-rank approximation of second and higher-order field interactions.
2. We formally prove the inference complexity of tensorFM and its direct predecessors. The inference complexity of tensorFM is $O(nkrd^2)$, where $n$ is the number of fields, $k$ is the size of feature embeddings, $d$ is the highest order of interactions used, and $r$ is the rank used for the approximation of the higher-order field interactions. This is potentially much lower than the $O(n^2k)$ complexity of FwFM, particularly when the number of fields is large, or more precisely when $n > rd^2$. However, our model allows the possibility of modeling higher-order interactions.
3. We provide an empirical comparison study between the new algorithm and several baseline models on three benchmark datasets.

The paper is organized as follows. Section 2 introduces the notation and formally discusses the direct predecessors to our proposed model. In Section 3, we derive the tensorFM model. Section 4 discusses the relation of our model to existing methods in the literature. In Section 5, we report the findings of our experiments on real and synthetic data.

## 2 Preliminaries

We introduce the notation used in the paper. Scalars are denoted by lowercase letters (e.g., $x$), column vectors by bold lowercase letters (e.g., $\boldsymbol{x}$), matrices by bold italic uppercase letters (e.g., $\boldsymbol{X}$), and tensors by bold uppercase letters (e.g., $\mathbf{X}$). The transpose of a column vector $\boldsymbol{x}$ is denoted with $\boldsymbol{x}^T$. Given a matrix $\boldsymbol{X} \in \mathbb{R}^{p \times q}$, we denote with the corresponding bold lowercase letter $\boldsymbol{x}_i \in \mathbb{R}^p$ the $i$-th column of $X$, where $1 \le i \le q$. We use $\langle \cdot, \cdot \rangle$ to denote the inner product between two vectors. Given two tensors $\mathbf{X}, \mathbf{Y} \in \mathbb{R}^{p_1 \times \dots \times p_\ell}$, we denote with $\mathbf{X} \circ \mathbf{Y}$ their Hadamard Product, we define their Frobenius inner product as $\langle \mathbf{X}, \mathbf{Y} \rangle_F = \sum_{i_1, \dots, i_\ell} \mathbf{X}_{i_1, \dots, i_\ell} \cdot \mathbf{Y}_{i_1, \dots, i_\ell}$, and denote by $\|\cdot\|_F$ the Frobenius norm induced by this inner product. For $\ell \ge 1$ and vectors $\boldsymbol{x}_1, \dots, \boldsymbol{x}_\ell$, their Kronecker (outer) product is the order-$\ell$ tensor $\boldsymbol{x}_1 \otimes \dots \otimes \boldsymbol{x}_\ell$ with entries $(\boldsymbol{x}_1 \otimes \dots \otimes \boldsymbol{x}_\ell)_{j_1, \dots, j_\ell} = \prod_{t=1}^{\ell} x_{t, j_t}$.

**Setting.** We consider a supervised learning setting over categorical data. In particular, we assume data composed of $n \ge 1$ categorical fields, where each field represents a specific attribute (e.g., the nationality of a user). Each field $j$, with $1 \le j \le n$, can take one of $m_j$ possible categorical values. We represent each input as a binary vector obtained by concatenating the one-hot encodings of the $n$ fields. Formally, let $\mathcal{D}_{m,n} = \{0, 1\}^m$ with $m = \sum_{j=1}^n m_j$ be our input domain, where each data point $\boldsymbol{x} \in \mathcal{D}_{m,n}$ is a binary feature vector, and we say that the data point contains feature $i$ if and only if $x_i = 1$. Each data point $\boldsymbol{x} \in \mathcal{D}_{m,n}$ has exactly $n$ features, i.e., $\|\boldsymbol{x}\|_1 = n$, one feature for each field, i.e., $\|\boldsymbol{x}\|_1 = n$.

The presentation in this work primarily focuses on binary classification tasks, where each input data point is associated with a binary label. We consider prediction functions of the form $\sigma \circ f : \mathcal{D}_{m,n} \to [0, 1]$, where $\sigma(x) = 1/(1 + e^{-x})$ denotes the sigmoid function and $f : \mathcal{D}_{m,n} \to \mathbb{R}$ is a scoring model. While the sigmoid transformation is natural for classification, the underlying framework and proposed models directly extend to other supervised tasks, such as regression, where the sigmoid transformation is unnecessary.

Given a model $f : \mathcal{D}_{m,n} \to \mathbb{R}$, we say that $f$ has inference complexity $t$ if there exists a procedure that evaluates $f(\boldsymbol{x})$ for any given $\boldsymbol{x} \in \mathcal{D}_{m,n}$ with computation complexity in $O(t)$. The goal is to design a scoring model $f$ that is accurate and has small inference complexity. The challenge of this problem is that the input $\boldsymbol{x}$ is high-dimensional and sparse. In our work, we are interested in a regime where both $m$ and $n$ are large.

**First-order interactions.** The simplest model is a linear scoring model

$$f_{\text{lin}}(\boldsymbol{x}) = \langle \boldsymbol{w}, \boldsymbol{x} \rangle + b \;, \tag{1}$$

where $\boldsymbol{w} \in \mathbb{R}^m$ and $b \in \mathbb{R}$ are parameters of the model. The resulting prediction function is also known as logistic regression (Cramer, 2002), and it can be computed in time $O(n)$ for any sparse input $\boldsymbol{x} \in \mathcal{D}_{m,n}$. This model only captures first-order interactions between features, as it evaluates the score based only on what features are present in the input $\boldsymbol{x}$ independently from the others.

**Second-order interactions.** The evaluation of cross-interactions between features is crucial to obtain more accurate predictions. Naively, those second-order interactions can be expressed by adding the score

$$f_{\text{second}}(\boldsymbol{x}) = \langle \boldsymbol{x}\boldsymbol{x}^T, \boldsymbol{W} \rangle_F = \sum_{i,j} x_i x_j \boldsymbol{W}_{i,j} \;, \tag{2}$$

where $\boldsymbol{W} \in \mathbb{R}^{m \times m}$ is a parameter describing the weights of a second degree polynomial. The computational time to evaluate equation 2 is $O(n^2)$, and a naive implementation of those second-order interactions presents two related shortcomings. First, an efficient evaluation of equation 2 would require storing the $m \times m$ matrix in primary memory, which is unfeasible for large $m$. Also, the matrix $\boldsymbol{W}$ has $O(m^2)$ parameters; thus, a large number of samples are required to obtain a statistically significant solution that generalizes.

There are two mainstream solutions to approach this problem. The first one is to hash the $O(m^2)$ pairs of indexes $(i, j) \in \{1, \dots, m\}^2$ into $\ell \ll m^2$ buckets, and associate each bucket with a single weight parameter. In this case, the computational complexity is still $O(n^2)$, however, the memory and the number of parameters is $O(\ell)$. In a seminal work, Rendle (2010) proposes the Factorization Machine (FM) model as another solution to tackle this problem.

In the FM model, feature $i$, with $1 \leq i \leq m$, is associated with a vector $\boldsymbol{v}_i \in \mathbb{R}^k$ of size $k$. The vectors $\{\boldsymbol{v}_i\}_{i=1}^m$ are also referred to as *embedding vectors*, and they are parameters of the model. Given an input $\boldsymbol{x} \in \mathcal{D}_{m,n}$, we denote with

$$\boldsymbol{A}_{\boldsymbol{x}} = [\boldsymbol{a}_{\boldsymbol{x},1}|\ldots|\boldsymbol{a}_{\boldsymbol{x},n}] \in \mathbb{R}^{k \times n} \tag{3}$$

the *embedding matrix* of $\boldsymbol{x}$, where the column $\boldsymbol{a}_{\boldsymbol{x},j}$ is the embedding vector $\boldsymbol{v}_i$ of the active feature, i.e., $x_i = 1$, associated with the field $j$. The FM model proposes to evaluate the second-order interactions by computing

$$f_{FM}(\boldsymbol{x}) = \sum_{i=1}^m \sum_{j=i+1}^m x_i x_j \langle \boldsymbol{v}_i, \boldsymbol{v}_j \rangle = \frac{1}{2} \langle \boldsymbol{A}_{\boldsymbol{x}}^T \boldsymbol{A}_{\boldsymbol{x}}, \boldsymbol{1}_n - \boldsymbol{I}_n \rangle_F \ ,$$

where $\boldsymbol{1}_n$ is a $n \times n$ matrix composed by only 1, and $\boldsymbol{I}_n$ is a $n \times n$ identity matrix. As an alternative viewpoint, it is easy to see that $f_{FM}(\boldsymbol{x}) = \frac{1}{2} \langle \boldsymbol{x}\boldsymbol{x}^T, (\boldsymbol{V}^T \boldsymbol{V}) \circ (\boldsymbol{1}_n - \boldsymbol{I}_n) \rangle_F$, where $\boldsymbol{V} \in \mathbb{R}^{k \times n}$ is the matrix whose columns are the embedding vectors of the features. This view illustrates that the FM model provides a factorization of the full $m \times m$ weight matrix $\boldsymbol{W}$ introduced in Equation 2 in terms of the embedding vectors $\boldsymbol{V}$, hence its name. FM has $O(mk)$ parameters, and it has inference complexity $O(nk)$.

Building on the idea of factorizing the coefficients of higher order interactions, by associating each feature with an embedding vector, a lot of different prediction models have been proposed throughout the last decade (e.g., Juan et al., 2017; Guo et al., 2017; Sun et al., 2021). Relevant to our work is the Field-weighted Factorization Machine model (FwFM) (Pan et al., 2018). Compared to the original FM model, the FwFM model has additional parameters $\boldsymbol{S} \in \mathbb{R}^{n \times n}$ that express the interaction strength between fields, i.e., the level of correlation between a field pair and the label. The intuition is that we would like to weigh more those cross-interactions between features of fields that have a stronger predictive power. The model can be expressed as follows:

$$f_{\text{FwFM}}(\boldsymbol{x}) = \frac{1}{2} \langle \boldsymbol{A}_{\boldsymbol{x}}^T \boldsymbol{A}_{\boldsymbol{x}}, \boldsymbol{S} \rangle \ , \tag{4}$$

where $\boldsymbol{S}$ is optimized in the space of symmetric matrices with zero diagonal. The FwFM has higher expressivity than an FM model but requires $O(n^2)$ additional parameters, and the inference complexity also worsens to $O(n^2 k)$. The work of (Shtoff et al., 2024b) approximates the field interaction matrix $S$ using a low-rank decomposition of rank $\rho$ to improve the complexity to $O(\rho n k)$, and our work can be seen as a direct generalization of the idea to higher orders of field interaction tensors.

**Higher-order interactions.** The factorization machine model has been extended to take into account higher-order interactions (Rendle, 2012). Given $p$ vectors $\boldsymbol{v}_1, \ldots, \boldsymbol{v}_p \in \mathbb{R}^k$, we let their $p$-way inner product be $\langle \boldsymbol{v}_1, \ldots, \boldsymbol{v}_p \rangle \doteq \sum_{i=1}^k \prod_{j=1}^p v_{i,j}$. In the higher-order FM model (HOFM), the $d$-order interactions can be expressed by evaluating $f_{p-FM}(\boldsymbol{x}) = \sum_{i_1 < i_2 < \ldots < i_d} x_{i_1} \ldots x_{i_d} \langle \boldsymbol{v}_{i_1}, \ldots, \boldsymbol{v}_{i_d} \rangle$. Unfortunately, a naive evaluation of the above expression requires an exponential number of steps $\Omega(n^d)$. To address this issue, different works propose variants of this model based on the ANOVA kernel or the polynomial kernel that have inference complexity $O(dnk)$ (Blondel et al., 2016a;b). Additional methods are discussed in the related work.

## 3  tensorFM **algorithm**

We derive the new tensorFM algorithm building upon the models discussed in the previous section. As a warm-up, we first consider second-order field interactions, observing that imposing additional low-rank structure on the matrix representing these interactions reduces inference complexity. This observation motivates us to generalize the low-rank approach to tensors representing higher-order field interactions, which is the main contribution of this section.

### 3.1 Warmup: Low-Rank Second-Order interactions

As a warm-up, we first consider second-order interactions. The FM and FwFM models can both be written in the unified form:

$$f(\boldsymbol{x}) = \langle \boldsymbol{A}_{\boldsymbol{x}}^{\top} \boldsymbol{A}_{\boldsymbol{x}}, \boldsymbol{S} \rangle_F \ , \tag{5}$$

where $\boldsymbol{S} = \frac{1}{2}(\mathbf{1}_n - \boldsymbol{I}_n)$ is fixed for FM models, and $\boldsymbol{S}$ is a symmetric zero-diagonal parameter matrix for FwFM models. While both models follow the structure of Equation 5, their inference complexities differ significantly. FM models admit inference in time $O(nk)$, whereas FwFM models generally require $O(n^2 k)$ time. The quadratic dependence on the number of fields in FwFM arises from the absence of structure in $\boldsymbol{S}$.

A natural approach to reducing inference complexity is to impose low-rank structures, a principle already exploited in the original factorization machine model. In the case of second-order interactions, it is possible to evaluate Equation 5 efficiently as a function of the rank of $\boldsymbol{S}$. The following proposition makes this explicit. Similar results appear in prior work (Shtoff et al., 2024b), but we include the simple argument here for completeness.

**Proposition 1.** *Let $\boldsymbol{S} \in \mathbb{R}^{n \times n}$ be a rank $r$ matrix. Then, after an appropriate preprocessing step depending only on $\boldsymbol{S}$, it is possible to evaluate $\langle \boldsymbol{A}_{\boldsymbol{x}}^T \boldsymbol{A}_{\boldsymbol{x}}, \boldsymbol{S} \rangle_F$ in time $O(rnk)$ for any $\boldsymbol{x} \in \mathcal{D}_{m,n}$.*

*Proof.* Since $\boldsymbol{S}$ is a rank $r$ matrix, it can be factorized using the singular value decomposition into three matrices $\boldsymbol{U}', \boldsymbol{V}' \in \mathbb{R}^{n \times r}$ and $\boldsymbol{D}' \in \mathbb{R}^{r \times r}$ such that $\boldsymbol{S} = \boldsymbol{U}' \boldsymbol{D}' \boldsymbol{V}'^T$. Let $\boldsymbol{U} = \boldsymbol{U}' \boldsymbol{D}'$, and $\boldsymbol{V} = \boldsymbol{V}'$, so that $\boldsymbol{S} = \boldsymbol{U} \boldsymbol{V}^T$, where $\boldsymbol{U}, \boldsymbol{V} \in \mathbb{R}^{n \times r}$. The following chain of inequality holds

$$\langle \boldsymbol{A}_{\boldsymbol{x}}^T \boldsymbol{A}_{\boldsymbol{x}}, \boldsymbol{S} \rangle_F = \langle \boldsymbol{A}_{\boldsymbol{x}}^T \boldsymbol{A}_{\boldsymbol{x}}, \boldsymbol{U} \boldsymbol{V}^T \rangle_F = \mathrm{Tr}(\boldsymbol{A}_{\boldsymbol{x}}^T \boldsymbol{A}_{\boldsymbol{x}} \boldsymbol{U} \boldsymbol{V}^T) = \mathrm{Tr}(\boldsymbol{V}^T \boldsymbol{A}_{\boldsymbol{x}}^T \boldsymbol{A}_{\boldsymbol{x}} \boldsymbol{U}) = \langle \boldsymbol{A}_{\boldsymbol{x}} \boldsymbol{V}, \boldsymbol{A}_{\boldsymbol{x}} \boldsymbol{U} \rangle_F \ ,$$

where the second and fourth equality are due to the relation between Frobenius inner product and trace, and the third equality is due to the cyclic property of the trace operator. The matrix multiplications $\boldsymbol{A}_{\boldsymbol{x}} \boldsymbol{U}$ and $\boldsymbol{A}_{\boldsymbol{x}} \boldsymbol{V}$ can be computed in time $O(rnk)$, and the statement immediately follows. $\square$

The inference complexity $O(nk)$ of the FM models follows as an immediate corollary of the above proposition. In fact,

$$f(\boldsymbol{x}) = \langle \boldsymbol{A}_{\boldsymbol{x}}^T \boldsymbol{A}_{\boldsymbol{x}}, \frac{1}{2}(\mathbf{1}_n - \boldsymbol{I}_n) \rangle_F = \frac{1}{2} \langle \boldsymbol{A}_{\boldsymbol{x}}^T \boldsymbol{A}_{\boldsymbol{x}}, \mathbf{1}_n \rangle_F - \frac{1}{2} \langle \boldsymbol{A}_{\boldsymbol{x}}^T \boldsymbol{A}_{\boldsymbol{x}}, \boldsymbol{I}_n \rangle_F \tag{6}$$

The second term of equation 6 can be computed in time $O(nk)$ as $\boldsymbol{I}_n$ has only $n$ non-zero entries, whereas the first term can be computed in time $O(nk)$ by noticing that $\mathbf{1}_n$ is a rank 1 matrix.

This discussion elucidates the following intuition. It is possible to obtain a model with low inference complexity that computes the second-order interactions as in equation 5 as long as the weights $\boldsymbol{S}$ have low rank. In our work, we build on these results and extend them to higher-order interactions.

### 3.2 Extension to Higher-Order Interactions

Let $\boldsymbol{A}_{\boldsymbol{x}} \in \mathbb{R}^{k \times n}$ be defined as in equation 3, where the columns of $\boldsymbol{A}_{\boldsymbol{x}}$ are the embedding vectors of the features in $\boldsymbol{x}$. We denote with $\overline{\boldsymbol{a}}_{\boldsymbol{x},1}, \ldots, \overline{\boldsymbol{a}}_{\boldsymbol{x},k}$ the $k$ rows of the matrix $\boldsymbol{A}_{\boldsymbol{x}}$.

Let $2 \leq d \leq n$ be the dimensionality of the order of interactions that we want to consider (e.g., $d = 2$ for second-order interactions). For a given $d$, our model considers $\ell$-order interactions for all $1 \leq \ell \leq d$. In particular, let $\mathsf{S}^{[\ell]} \in \mathbb{R}^{n \times \cdots \times n}$ be an $\ell$ order tensor for $2 \leq \ell \leq d$. In our work, we consider the following model:

$$f_{tFM}(\boldsymbol{x}) = f_{\mathrm{lin}}(\boldsymbol{x}) + \sum_{\ell=2}^{d} \sum_{i_1, \ldots, i_\ell} \mathsf{S}^{[\ell]}_{i_1, \ldots, i_\ell} \cdot \langle \boldsymbol{a}_{\boldsymbol{x}, i_1}, \ldots, \boldsymbol{a}_{\boldsymbol{x}, i_\ell} \rangle \tag{7}$$

Analogous to how the matrix $\boldsymbol{S}$ models second-order field interaction strengths in FwFM, here each tensor $\mathsf{S}^{[\ell]}$ models $\ell$-order field interaction strengths. Without any additional constraint, the inference complexity to

evaluate this model for any given $\boldsymbol{x} \in \mathcal{D}_{m,n}$ is equal to $O(d^2 k n^d)$, which is intractable even for small $d$. To reduce the inference complexity of the model, we build on the intuition developed in Section 3.1. In particular, we want to constrain a notion of rank of the tensor $\mathsf{S}^{[\ell]}$ so that it is possible to evaluate Equation 7 efficiently.

Given an $\ell$ order tensor $\mathsf{W} \in \mathbb{R}^{n \times \cdots \times n}$, we say that $\mathsf{W}$ has canonical polyadic (CP) rank at most $r$ if there exists a collection of $\ell$ matrices $\boldsymbol{U}^{[1]}, \ldots, \boldsymbol{U}^{[\ell]} \in \mathbb{R}^{n \times r}$ such that

$$\mathsf{W} = \sum_{i=1}^{r} \boldsymbol{u}_i^{[1]} \otimes \ldots \otimes \boldsymbol{u}_i^{[\ell]} \ . \tag{8}$$

The CP rank of $\mathsf{W}$ is the minimum $r \geq 0$ for which such a decomposition exists (Hitchcock, 1927; Kolda & Bader, 2009). Note that for $\ell = 2$, the CP rank coincides with the rank of the matrix $\boldsymbol{W}$, and the decomposition in equation 8 can be obtained through the singular value decomposition (i.e., $\boldsymbol{U}^{[1]} = \boldsymbol{U}$ and $\boldsymbol{U}^{[2]} = \boldsymbol{V}$, where $\boldsymbol{U}$ and $\boldsymbol{V}$ are defined as in the proof of Proposition 1). The following result shows that the CP rank can be used to upper bound the inference complexity of evaluating higher-order interactions, and it is a generalization of Proposition 1.

**Lemma 1.** *Let $\boldsymbol{A_x} \in \mathbb{R}^{k \times n}$. Let $\mathsf{W}^{[\ell]} \in \mathbb{R}^{n \times \cdots \times n}$ be a $\ell$-order tensor with CP rank $r$. Then, after an appropriate preprocessing step depending only on $\mathsf{W}^{[\ell]}$, it is possible to evaluate $T = \sum_{i_1, \ldots, i_\ell} \mathsf{W}_{i_1, \ldots, i_\ell}^{[\ell]} \cdot \langle \boldsymbol{a}_{\boldsymbol{x}, i_1}, \ldots, \boldsymbol{a}_{\boldsymbol{x}, i_\ell} \rangle$ for any given input $\boldsymbol{x}$ with time complexity $O(\ell r k n)$.*

*Proof.* The value $T$ can be rewritten as

$$T = \left\langle \sum_{i=1}^{k} \underbrace{\overline{\boldsymbol{a}}_{\boldsymbol{x}, i} \otimes \ldots \otimes \overline{\boldsymbol{a}}_{\boldsymbol{x}, i}}_{\ell \text{ times}}, \mathsf{W}^{[\ell]} \right\rangle_F \ .$$

Since $\mathsf{W}$ has CP rank $r$, there exists a collection of $\ell$ matrices $\boldsymbol{U}^{[1]}, \ldots, \boldsymbol{U}^{[\ell]}$ that satisfy Equation 8, thus:

$$T = \left\langle \sum_{i=1}^{k} \underbrace{\overline{\boldsymbol{a}}_{\boldsymbol{x}, i} \otimes \ldots \otimes \overline{\boldsymbol{a}}_{\boldsymbol{x}, i}}_{\ell \text{ times}}, \sum_{j=1}^{r} \boldsymbol{u}_j^{[1]} \otimes \ldots \otimes \boldsymbol{u}_j^{[\ell]} \right\rangle_F \ .$$

Since the inner product is a bilinear operator, we have that

$$T = \sum_{i=1}^{k} \sum_{j=1}^{r} \left\langle \underbrace{\overline{\boldsymbol{a}}_{\boldsymbol{x}, i} \otimes \ldots \otimes \overline{\boldsymbol{a}}_{\boldsymbol{x}, i}}_{\ell \text{ times}}, \boldsymbol{u}_j^{[1]} \otimes \ldots \otimes \boldsymbol{u}_j^{[\ell]} \right\rangle_F$$

If we expand the computations, we have that $T$ is equal to

$$T = \sum_{i=1}^{k} \sum_{j=1}^{r} \sum_{a_1, \ldots, a_\ell} \prod_{b=1}^{\ell} (u_{i, a_b}^{[b]} \overline{a}_{\boldsymbol{x}, j, a_b}) = \sum_{i=1}^{k} \sum_{j=1}^{r} \prod_{b=1}^{\ell} \left( \sum_{a=1}^{n} u_{i,a}^{[b]} \overline{a}_{\boldsymbol{x}, j, a} \right) = \sum_{i=1}^{k} \sum_{j=1}^{r} \prod_{b=1}^{\ell} \langle \boldsymbol{u}_j^{[b]}, \overline{\boldsymbol{a}}_{\boldsymbol{x}, i} \rangle \ ,$$

which can be computed in time $O(\ell n k r)$. $\qquad \square$

The CP decomposition of Equation 8 is not the only known tensor decomposition. In Appendix A, we explore a parallel to Lemma 1 based on another popular tensor decomposition strategy, known as higher-order singular value decomposition (HOSVD) or Tucker decomposition.

### 3.3 Low-rank higher-order interactions: tensorFM

Equipped with Lemma 1, we are ready to formally introduce the tensorFM model. As in a factorization model, we consider an embedding vector $\boldsymbol{v}_i \in \mathbb{R}^k$ for each feature $1 \leq i \leq n$. Our model is defined by parameters $d$ and a vector $\boldsymbol{r} = (r_2, \ldots, r_d)^T$, where $1 \leq r_\ell \leq n$ for any $2 \leq \ell \leq d$. For any $2 \leq \ell \leq d$, we have parameters $\boldsymbol{U}_1^{[\ell]}, \ldots, \boldsymbol{U}_\ell^{[\ell]} \in \mathbb{R}^{n \times r_\ell}$. Given those parameters, the tensor $\mathsf{S}^{[\ell]}$ is defined according to the decomposition

in Equation 8, i.e., $\mathsf{S}^{[\ell]} = \sum_{i=1}^{r_\ell} \boldsymbol{u}_{1,i}^{[\ell]} \otimes \ldots \otimes \boldsymbol{u}_{\ell,i}^{[\ell]}$. The tensorFM model is defined as in equation 7 using tensors $\mathsf{S}^{[2]}, \ldots, \mathsf{S}^{[\ell]}$. Given $d$ and $\boldsymbol{r}$, we define an instance of this model as tensorFM$(\boldsymbol{r}, d)$. This model has $O(mk + n\sum_{\ell=2}^{d} \ell r_\ell)$ parameters. The inference complexity of this model is $O(nk \sum_{\ell=2}^{d} \ell \cdot r_\ell)$ and follows by using Lemma 1.

**Theorem 1.** *Let $d \geq 2$ and $\boldsymbol{r} = (r_2, \ldots, r_d) \in \{1, \ldots, n\}^{d-1}$. The model tensorFM$(\boldsymbol{r}, d)$ has inference complexity $O(nk \sum_{\ell=2}^{d} \ell \cdot r_\ell)$.*

*Proof.* Fix an arbitrary $\boldsymbol{x} \in \mathcal{D}_{m,n}$. The linear component of $f_{lin}(\boldsymbol{x})$ can be computed in time $O(n)$. For any $2 \leq \ell \leq d$, we can compute $\sum_{i_1, \ldots, i_\ell} \mathsf{S}_{i_1, \ldots, i_\ell}^{[\ell]} \cdot \langle \boldsymbol{a}_{\boldsymbol{x}, i_1}, \ldots, \boldsymbol{a}_{\boldsymbol{x}, i_\ell} \rangle$ in time $O(\ell r_\ell kn)$ using Lemma 1. The statement follows by summing over all possible values of $\ell$. $\qquad\square$

*Remark 1.* This inference complexity can be expressed using the looser bound $O(nkrd^2)$, where $r$ is the highest rank in $\boldsymbol{r}$.

*Remark 2.* We do not impose any symmetry constraint on the tensors $\mathsf{S}^{[\ell]}$. Given any $\ell$-order tensor $\mathsf{S}^{[\ell]}$, consider its symmetrization

$$\hat{\mathsf{S}}_{i_1, \ldots, i_\ell}^{[\ell]} = \frac{1}{\ell!} \sum_{\pi \in \mathbb{S}_\ell} S_{i_{\pi(1)}, \ldots, i_{\pi(\ell)}}^{[\ell]},$$

where $\mathbb{S}_\ell$ denotes the set of all permutations $\pi$ of the index tuple $(i_1, \ldots, i_\ell)$. Because the multilinear form

$$\sum_{i_1, \ldots, i_\ell} \mathsf{S}_{i_1, \ldots, i_\ell}^{[\ell]} \langle \boldsymbol{a}_{\boldsymbol{x}, i_1}, \ldots, \boldsymbol{a}_{\boldsymbol{x}, i_\ell} \rangle$$

is invariant under such index permutations, the tensors $\mathsf{S}^{[\ell]}$ and $\hat{\mathsf{S}}^{[\ell]}$ yield the same output. Thus, a symmetry constraint would not reduce the expressivity of our model. On the other hand, our model *does* allow self-interactions (i.e., the multilinear form has index tuples with repeated coordinates), which marks a key difference from other models such as FwFM.

**Comparison with HOFM**. A key distinction between our approach and higher-order factorization machines (HOFM, Rendle (2012)) lies in the treatment of the interaction tensors $\mathsf{S}^{[\ell]}$ in Equation 7. In HOFM, $\mathsf{S}^{[\ell]}$ is fixed a priori and chosen to allow efficient computation of the resulting polynomial via mathematical identities such as Newton's identities. This is analogous to how the original FM model fixes its interaction matrix.

In contrast, our framework aims to learn the tensor $\mathsf{S}^{[\ell]}$ for each order $\ell$, while still maintaining computational efficiency. This is achieved by constraining $\mathsf{S}^{[\ell]}$ to have low CP rank, which ensures that the evaluation of the model remains tractable even when the interaction structure is learned from data.

**Numerical Features.** While the focus is on categorical features, our method (like other factorization machine-based approaches) naturally extends to numerical features. When $x_i \in \mathbb{R}$ is a numerical feature, we add the corresponding column $x_i \boldsymbol{v}_i$ to $\boldsymbol{A}_{\boldsymbol{x}}$, where $\boldsymbol{v}_i$ is the embedding vector associated with that feature. Recent works have proposed alternative embeddings for numerical features that are compatible with our setting, where each field is associated with a single embedding vector (Gorishniy et al., 2022; Shtoff et al., 2024a; Rügamer, 2024).

## 4 Related Work

For $d = 2$, the second-order interactions of tensorFM$(d, r)$ model coincide with equation 5, where $S$ is parameterized as the product $S = U_1^{[2]} U_2^{[2]T}$ of two rectangular matrices. In the optimization literature, this decomposition is common to optimize a low-rank matrix. Thus, for $d = 2$, our model can be seen as a modified version of FwFM that allows a trade-off between the number of parameters $O(nr)$ and the inference complexity $O(nkr)$ that is regulated by the rank parameter $r$ (these values coincide with the ones of the FwFM models for $r = n$). The idea of using low-rank matrix decompositions has also been applied in recent variants of factorization machines for the special case of $d = 2$. Almagor & Hoshen (2022) propose a new method where each field $i$, with $1 \leq i \leq n$ is associated with a parameter $U_i \in \mathbb{R}^{k \times r}$, where $r$ is a rank parameter.

The cross-interaction between the features of fields $i$ and $j$ is computed as $\boldsymbol{v}_i^T \boldsymbol{M}_{ij} \boldsymbol{v}_j$, where $M_{ij} = \boldsymbol{U}_i \boldsymbol{\Lambda} \boldsymbol{U}_j^T$, and $\boldsymbol{\Lambda} \in \mathbb{R}^{k \times k}$ is another parameter. While this work shares similar ideas, their approach is limited to second-order interactions and does not straightforwardly extend to higher-order interactions. A similar low-rank decomposition of the field-interaction matrix has also been proposed in other recent work (Shtoff et al., 2024b), where the authors introduce the interaction matrix $\boldsymbol{S} = \boldsymbol{U}\Lambda \boldsymbol{U}^\top - \left[ (\boldsymbol{U}\boldsymbol{\Lambda}\boldsymbol{U}^\top) \odot \boldsymbol{I}_n \right]$, with $\boldsymbol{U} \in \mathbb{R}^{n \times r}$, $\boldsymbol{\Lambda} \in \mathbb{R}^{r \times r}$ is diagonal, and $\odot$ denotes the element-wise (Hadamard) product. This construction is closely related to our model, with the additional modeling constraint that $S$ has zero-diagonal. Our model generalizes the ideas of the aforementioned papers beyond second-order interactions ($d = 2$) and can thus capture higher-order interactions.

For $d > 2$, the tensorFM model considers higher-order interactions between features' embeddings. The first HOFM was proposed in the original work on FM (Rendle, 2010; 2012), and it considers a fixed and given weighting of the higher-order interactions, which is equivalent to our model equation 7 with fixed tensors $\mathsf{S}^{[2]}, \ldots, \mathsf{S}^{[\ell]}$. The variants of HOFM that consider the ANOVA kernel (Blondel et al., 2016a) and the polynomial kernel (Blondel et al., 2016b) can be seen as a special selection of $\mathsf{S}^{[2]}, \ldots, \mathsf{S}^{[\ell]}$ that allows a fast computation of the model. Conversely, in our work, rather than considering a fixed weighting of the higher-order interactions between feature's embeddings of the $n$ fields, we consider a more expressive model that can learn the tensors $\mathsf{S}^{[2]}, \ldots, \mathsf{S}^{[\ell]}$, while guaranteeing that the resulting model with this weighting can be still computed efficiently. Our method can be seen as a generalization of the FwFM model that enables us to both improve the inference complexity and consider higher-order interactions.

The aforementioned models and our work explicitly compute a polynomial over the cross-products between feature embeddings. In the literature, many other models have been proposed that compute higher-order interactions through specific neural network architectures.

The attention-aware factorization machine is another factorization machine model that considers a polynomial over second-order interactions as in equation 5, however the weights $\boldsymbol{S}$ are chosen depending on $\boldsymbol{A_x}$ through a neural attention network (Xiao et al., 2017). This architecture requires the computation of all the dot products between the pairs of vectors' embeddings of the features of $\boldsymbol{x}$, thus the inference complexity is $\Omega(n^2 k)$. Another model called AutoInt automatically learns high-order feature interactions through a multi-head self-attentive neural network, but its inference complexity is still $\Omega(n^2)$.

The Cross Interaction Network proposes a network model that can provably approximate a special class of $\ell$ degree polynomial over cross-interactions between embedding vectors (Lian et al., 2018). However, their analysis of this polynomial approximation uses a version of this model for which the inference complexity is $O(n^3 k \ell)$, which is cubic in $n$.

Given $\boldsymbol{x}$, the cross-networks receive in input the concatenation of the vectors $\boldsymbol{y} = (\boldsymbol{a_{x,1}}, \ldots, \boldsymbol{a_{x,n}}) \in \mathbb{R}^{n \cdot k}$, and they iteratively transform $\boldsymbol{y}$ with a forward process for $\ell$ steps, where $\ell$ is the number of layers (Wang et al., 2017; 2021). It is possible to show that the CN networks express a special class of $\ell + 1$ degree polynomials over the components $y_1, \ldots, y_{nk}$. Compared to our method, the cross-interactions are measured at the bit level (using the vector $\boldsymbol{y}$), rather than at the feature level. Our model has the advantage of being interpretable, as it explicitly learns the interaction strength of cross-interactions between different fields.

**Deep Networks.** In recent work, it has been shown that a properly fine-tuned Deep Neural Network (DNN) with a large enough layer size can be competitive with many baselines for recommender systems. In the literature, DNNs are often trained in parallel with a simpler model that captures lower-order interaction (e.g., FM, CN, CIN), i.e., the resulting model is $f(\boldsymbol{x}) = g(f_{\text{simple}}(\boldsymbol{x}), f_{\text{DNN}}(\boldsymbol{x}))$ with $g(\cdot, \cdot)$ being some "aggregation" function of two models, for example, a sum. DNN networks are a class of universal function approximators Hornik et al. (1989) and can capture implicitly higher-order feature interactions, while the simpler model explicitly models the lower-level interaction. It has been shown in a series of works that the combination of these two architectures leads to more accurate models (e.g, Guo et al., 2017; Wang et al., 2017; LeGendre et al., 2019; Wang et al., 2021). Since we are interested in developing a simple model that can capture cross-order interactions between features with low inference delay, we do not employ a DNN component which would significantly increase the inference complexity. We remark that deep networks have been widely used for recommendation systems, and we refer to a comprehensive survey by Zhang et al. (2019).

## 5   Experiments

The empirical studies compare the introduced tensorFM model with state-of-the-art competitors on three benchmark datasets (Section 5.1 and Section 5.2). We also include an analysis of the inference time (Section 5.5). The code for the experiments is provided in the supplementary material.

**Datasets.** We evaluate our proposed model tensorFM on three open benchmark datasets `Avazu` (Avazu, 2014), `Criteo` Labs (2014), and `COMPAS` (Washington, 2018). The `Avazu` and `Criteo` datasets contain online advertising click-through data used for predicting click-through rates. The `COMPAS` dataset contains criminal justice records with demographic and offense-related features, used for predicting recidivism within two years. Table 3 summarizes key information about those datasets.

**Baselines.** The goal of our work is to develop a Factorization Machine model that explicitly considers cross-interactions between features and exhibits low inference complexity. Consequently, we compare our model with other baselines that have relatively low inference complexity, excluding deep architectures that exhibit significantly larger inference complexity. We use the following baselines:

1. **LR**: Logistic Regression,
2. **FM**: Factorization Machine (Rendle, 2010),
3. **HOFM**: Higher-Order Factorization Machine (Blondel et al., 2016a)
4. **FwFM**: Field-weighted Factorization Machines (Pan et al., 2018),
5. **AFM**: Attentional Factorization Machines (Xiao et al., 2017), with 8 attention heads,
6. **CN**: Cross Network (Wang et al., 2021), with two layers and a vector to parameterize the CN as in (Wang et al., 2017).

Noticeably, we excluded CIN and AutoINT, as they exhibit a considerably larger inference time (see discussion in Section 5.5).

We parameterize our model with two integers $r$ and $d$, and we let tensorFM$(r, d)$ be equal to the model as defined in Section 3 where we fix the rank $r$ for each higher-order interaction, i.e., $\boldsymbol{r} = (r, \ldots, r)$. For all baseline models, we use the implementation from the pytorch-fm package[1].

**Hyperparameters.** The embedding size is fixed to $k = 8$ throughout all the experiments. We perform a hyper-parameter search using a validation dataset. For the hyper-parameter tuning, we adopt the open source hyper-parameter optimizer Optuna (Akiba et al., 2019) for 50 steps, using AUC as optimization target. The learning rate is chosen in the interval $[1e - 4, 0.1]$ and the regularization coefficients in the interval $[1e - 4, 0]$. All the methods are trained for 5 epochs with a batch size equal to 1024.

For our method, we try low-rank configurations tensorFM$(r, d)$, where $r \in \{1, 2, 4\}$ and $d \in \{2, 3, 4\}$ that exhibit low inference complexity. We refer to those models as *low-complexity tensorFM*. We also report results for *high-complexity tensorFM* where $r \in \{8, 16, 32\}$. We adopt AdaGrad as optimization algorithm, and use binary cross-entropy loss.

### 5.1   Online Advertising Benchmark

In this subsection, we report the experimental results for the datasets `Criteo` and `Avazu`. In Table 2, we report the test performance of the different models evaluated based on the AUC metric (Area Under the ROC curve). In the table, we report the best-performing low-complexity tensorFM and high-complexity tensorFM models (chosen using the validation set). For `Avazu`, these are respectively tensorFM$(4, 4)$ and tensorFM$(4, 16)$, and for `Criteo`, these are tensorFM$(3, 4)$ and tensorFM$(3, 32)$.

Our method obtains competitive performance with respect to the existing baselines. In particular, the high-complexity tensorFM model achieves top two AUC on both datasets. The low-complexity tensorFM also obtains competitive results. If we compare low-complexity tensorFM to FwFM, our model has fewer parameters and a smaller inference complexity. However, the performance of our model matches or outperforms the FwFM model. This suggests that taking into account higher-order interactions can improve the prediction performance.

---

[1] https://github.com/rixwew/pytorch-fm

Table 2: Comparison of baseline performance in terms of the test set AUC (%) and LogLoss. The best results are **bold** and the second best results are underlined.

| Model | Avazu | | Criteo | |
|---|---|---|---|---|
| | Test Log-Loss | Test AUC | Test Log-Loss | Test AUC |
| LR | 0.3873 | 76.53 | 0.4551 | 79.39 |
| FM | 0.3810 | 77.69 | 0.4470 | 80.44 |
| FwFM | 0.3805 | 77.28 | 0.4428 | 80.87 |
| AFM | 0.3833 | 77.31 | 0.4462 | 79.75 |
| CN | 0.3841 | 77.16 | 0.4514 | 79.55 |
| HOFM | **0.3797** | **77.91** | 0.4449 | 80.67 |
| low-complexity tensorFM | 0.3805 | 77.74 | 0.4436 | 80.79 |
| high-complexity tensorFM | 0.3804 | 77.77 | **0.4422** | **80.94** |

Table 3: Dataset statistics.

| | Avazu | Criteo | COMPAS |
|---|---|---|---|
| Train set size | 28,300,276 | 33,003,326 | 4098 |
| Valid. size | 4,042,897 | 8,250,124 | 879 |
| Test size | 8,085,794 | 4,587,167 | 879 |
| # features ($m$) | 1,544,250 | 2,086,936 | 204 |
| # fields ($n$) | 22 | 39 | 14 |

Table 4: Recidivism based on COMPAS dataset.

| Baselines | AUC | TensorFM | AUC |
|---|---|---|---|
| LR | 83.87 | tensorFM(1,2) | 84.87 |
| FM | 84.03 | tensorFM(2,2) | 84.96 |
| FwFM | 84.24 | tensorFM(1,3) | **85.29** |
| AFM | 84.13 | tensorFM(2,3) | 85.02 |
| CN | 84.33 | tensorFM(1,4) | 84.44 |

## 5.2 COMPAS Dataset

The COMPAS (Correctional Offender Management Profiling for Alternative Sanctions) dataset (Washington, 2018) consists of data from individuals involved in the U.S. criminal justice system. Each record contains 14 fields, including demographic information (such as age, race, and sex) and details of interactions with the criminal justice system (such as criminal history, offense date, and time spent in jail). The dataset also includes a binary recidivism outcome, indicating whether an individual re-offended within two years. This dataset allows for the training of machine learning models to predict the likelihood that an individual will commit another crime within the next two years.

We normalize the numerical features to fall within the range [0, 1] and further discretize them into 5 bins. The dataset is split into training, validation, and test sets, with proportions of 70%, 15%, and 15%, respectively. The results are presented in Table 4. As we can see, the low-complexity tensorFM models are able to achieve competitive (or even better) results compared to existing baselines.

## 5.3 Synthetic Data

In this subsection, we use synthetic data to verify that tensorFM effectively captures higher-order interactions. The dataset is generated as follows: we consider data points with three fields, each taking 20 possible values. Each triplet of features (one from each field) is assigned a random binary label uniformly at random. The dataset consists of one million data points, sampled uniformly over all possible triplets of features.

By construction, the label of each data point is entirely determined by the triplet of its features, meaning that classification relies on third-order interactions. Table 5 reports the results of our experiments on a randomly generated dataset as described above. As expected, logistic regression, a linear model, performs poorly. FM and FwFM perform better, as they can capture second-order interactions, but they exhibit lower performance than AFM and tensorFM, which are capable of capturing higher-order interactions.

Similarly, we test same baselines on the fourth-order interactions data. In this case, we consider four fields and every four-tuple of features (one from each field) is assigned a random binary label uniformly at random.

Table 5: Comparison of baseline performance in terms of the test set AUC and LogLoss (%) over the synthetic data for 3-wise and 4-wise interactions. The best results are **bold** and the second best results are underlined.

| Model | 3-wise Interactions | | 4-wise Interactions | |
|---|---|---|---|---|
| | Test Log-Loss | Test AUC | Test Log-Loss | Test AUC |
| LR | 0.6896 | 54.87 | 0.6917 | 53.18 |
| FM | 0.6508 | 66.07 | 0.6738 | 61.11 |
| FwFM | 0.6514 | 65.99 | 0.6734 | 61.07 |
| AFM | **0.6218** | **71.96** | 0.6730 | 61.56 |
| CN | 0.6676 | 61.73 | 0.6813 | 57.70 |
| tensorFM(3,3) | 0.6239 | 70.43 | **0.6583** | **64.68** |

In Table 6, we additionally evaluate a more challenging synthetic setup, where the original 4-wise interaction data is augmented with 96 noise fields (for a total of 100 fields). The results confirm that tensorFM more effectively recovers the underlying fourth-order interactions compared to the baselines, even under substantial noise and increased dimensional complexity.

Table 6: Comparison of baseline performance in terms of the test set AUC (%) and LogLoss over the synthetic data containing 100 columns, for 3-wise and 4-wise interactions. The best results are **bold** and the second best results are underlined.

| Model | 3-wise Interactions | | 4-wise Interactions | |
|---|---|---|---|---|
| | Test Log-Loss | Test AUC | Test Log-Loss | Test AUC |
| LR | 0.6912 | 53.59 | 0.6896 | 54.87 |
| FM | 0.6869 | 58.01 | 0.6883 | 55.83 |
| FwFM | 0.6572 | 64.99 | 0.6771 | 60.18 |
| AFM | 0.6912 | 53.59 | 0.6914 | 53.36 |
| CN | 0.6834 | 57.76 | 0.6867 | 56.04 |
| HOFM(3) | 0.6913 | 55.87 | 0.6892 | 55.20 |
| tensorFM(3,3) | **0.6325** | **68.91** | 0.6618 | 63.85 |
| tensorFM(4,4) | 0.6331 | 68.53 | **0.6526** | **65.55** |

## 5.4 Ablation Study

In this section, we examine how the rank $r$ and the interaction order $d$ affect the performance of our method, tensorFM. The experiments are conducted on the `Avazu` dataset. Figure 1 shows the test AUC as a function of the rank $r$. We observe that increasing the rank leads to improved model accuracy. Additionally, higher-order interactions also contribute to better performance, indicating that both rank and interaction order play important roles in enhancing the model's accuracy. Figure 2 also implies the binary cross entropy loss decreases as we increase rank and the order of interaction.

## 5.5 Inference Time

In this subsection, we measure the inference complexity using an estimate of the number of Floating Point Operations (FLOPs) necessary to compute the label of a given vector $\boldsymbol{x} \in \mathcal{D}_{m,n}$.

This comparison follows the example of previous work (Sun et al., 2021), and allows us to compare the inference complexity regardless of the implementation details. We compare the number of FLOPs required for inference for both our model and the baselines as a function of the number of fields $n$ (i.e., the number of

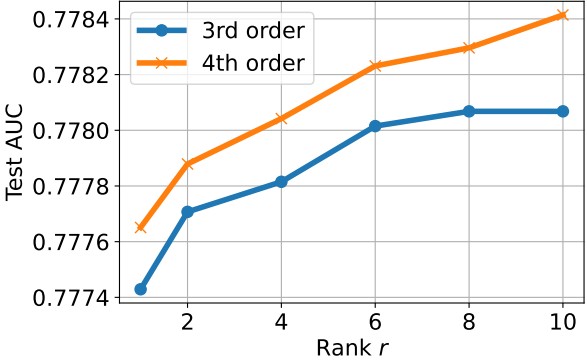

Figure 1: Test AUC as a function of rank $r$. Increasing rank and order of interaction improves the performance.

Figure 2: Loss as a function of rank $r$. Increasing rank and order of interaction decreases the loss.

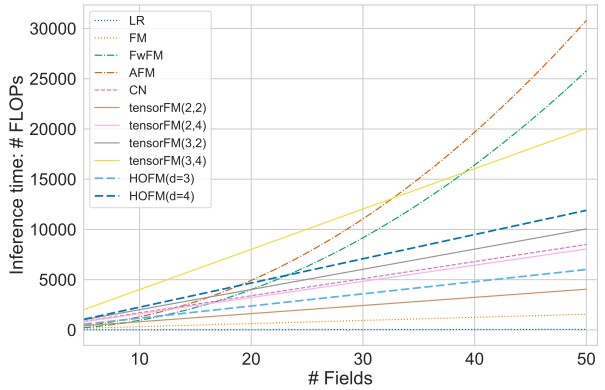

Figure 3: Inference time of different models measured in FLOPs varying the number of input fields.

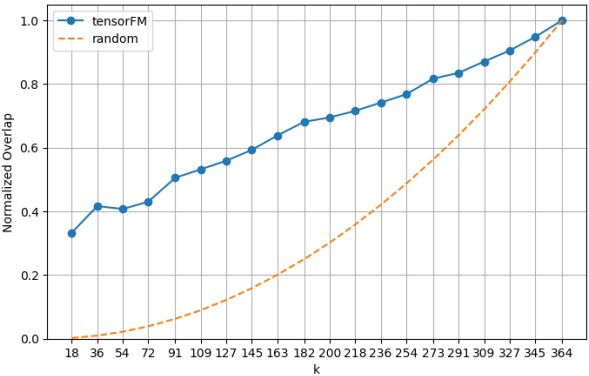

Figure 4: Overlap of top-$k$ interactions: tensorFM(3,3) vs mutual information. The dashed line represents the expected overlap with a random order.

non-zero entries of $\boldsymbol{x}$). The results are reported in Figure 3. We observe that the number of FLOPs scales linearly with the number of fields $n$ for our model tensorFM, as well as for LR, FM, and CN.

In contrast, AFM and FwFM demonstrate a quadratic increase in FLOPs counts with respect to $n$, leading to a substantially higher number of FLOPs for larger inputs. The FLOPs counts for CIN and AutoINT, also proportional to $\Omega(n^2)$, were omitted from the plot due to their significantly larger values (see Table 7 for a comprehensive details of the inference complexity for different models). The fastest models, LR and FM, demonstrate lower accuracy as detailed in Table 2. In contrast, our method not only achieves competitive accuracy relative to the other baselines but also exhibits a low inference time that scales linearly with the number of fields $n$.

We conduct an additional set of experiments to evaluate inference time at serving using an optimized Cython implementation. Specifically, we compare the runtime of FwFM, CN, AFM, HOFM, and TensorFM. Similarly, we exclude LR and FM, as they are linear model with substantially lower inference times. Our evaluation is performed on input points with $n = 100$ fields and an embedding size $k = 8$. We report the average inference time over 100000 points in Table 7.

## 5.6 Interpretability

Our model enables interpretability by explicitly parameterizing $\ell$-way feature interactions. In particular, the interaction tensor $\mathsf{S}^{[\ell]}$ can be directly computed from the learned parameters using the decomposition

Table 7: Inference complexity (right) and empirical inference time in milliseconds (left) of different models. The empirical inference complexity is evaluated as the average inference time on 100000 synthetic data points with $n = 100$ fields and embedding size $k = 8$.

| Model | Time (ms) |
|---|---|
| TensorFM(1,2) | 0.00115 |
| TensorFM(2,2) | 0.00152 |
| TensorFM(4,2) | 0.00286 |
| TensorFM(1,3) | 0.00194 |
| TensorFM(2,3) | 0.00367 |
| TensorFM(4,3) | 0.00703 |
| TensorFM(1,4) | 0.00340 |
| TensorFM(2,4) | 0.00653 |
| TensorFM(4,4) | 0.01292 |
| FwFM | 0.01265 |
| CN | 0.00456 |
| AFM | 0.18366 |
| HOFM | 0.00488 |

| Model | Inference Complexity |
|---|---|
| LR | $O(n)$ |
| FM | $O(nk)$ |
| FwFM | $O(n^2 k)$ |
| AFM | $O(n^2 k)$ |
| CN(layer) | $O(nk \cdot \texttt{\#layers})$ |
| AutoINT | $O(n^2)$ |
| CIN | $\Omega(n^3)$ |
| HOFM(d) | $O(nkd)$ |
| TensorFM(d,r) | $O(nkrd^2)$ |

in Equation 8. We remind that this tensor encodes the strength of $\ell$-way interactions among fields. To evaluate how well these model-induced interactions reflect true statistical dependencies, we compare the learned interaction values with mutual information computed on the training set.

For any tuple of $\ell$ features $(i_1, \dots, i_\ell)$ with feature $i_r$ belonging to field $F_{k_r}$ for $1 \leq r \leq \ell$, we let

$$I(i_1, \dots, i_\ell) = \mathbf{S}^{[\ell]}_{k_1, \dots, k_\ell} \cdot \langle a_{i_1}, \dots, a_{i_\ell} \rangle,$$

where $a_i$ is the embedding vector associated with feature $i$. We define the learned interaction strength between fields $(F_{k_1}, \dots, F_{k_\ell})$ as

$$\frac{\sum_{(i_1, \dots, i_\ell) \in (F_{k_1}, \dots, F_{k_\ell})} |I(i_1, \dots, i_\ell)| \cdot \#(i_1, \dots, i_\ell)}{\sum_{(i_1, \dots, i_\ell) \in (F_{k_1}, \dots, F_{k_\ell})} \#(i_1, \dots, i_\ell)},$$

where $\#(i_1, \dots, i_\ell)$ denotes the number of times the feature tuple appears in the training set. We compare this with the mutual information between $\ell$-tuples of fields and the label, estimated from the empirical distribution $p$ on the training set:

$$MI((F_{k_1}, \dots, F_{k_\ell}); Y) = \sum_{(i_1, \dots, i_\ell) \in (F_{k_1}, \dots, F_{k_\ell})} \sum_{y \in Y} p((i_1, \dots, i_\ell), y) \log \left( \frac{p((i_1, \dots, i_\ell), y)}{p(i_1, \dots, i_\ell) p(y)} \right) \quad .$$

where the sum is over tuples with $i_r \in F_{k_r}$.

We perform this analysis on the COMPAS dataset using a `tensorFM(3,3)` model. We observe a Pearson correlation of 0.3995 between the learned interaction strengths and the mutual information values, computed across all possible $\ell$-tuples of fields. In addition to correlation, we measure the overlap between the top-$k$ most strongly interacting field combinations under both metrics. As shown in Figure 4, the blue line represents the overlap between the top-$k$ triplets ranked by learned interaction strength and those ranked by mutual information. The observed intersection rate is consistently higher than the expected overlap $(k/n)^2$ under a random ordering (dashed orange line), indicating a strong alignment between model-induced and data-driven interactions. In particular, for the top-36 triplets, the overlap exceeds 40%.

A key advantage of tensorFM is its ability to make higher-order feature interactions directly interpretable. In particular, it is possible to explicitly visualize the significant learned field interactions. Figure 5 shows a heatmap of the 36 most significant learned third-order interaction strengths.

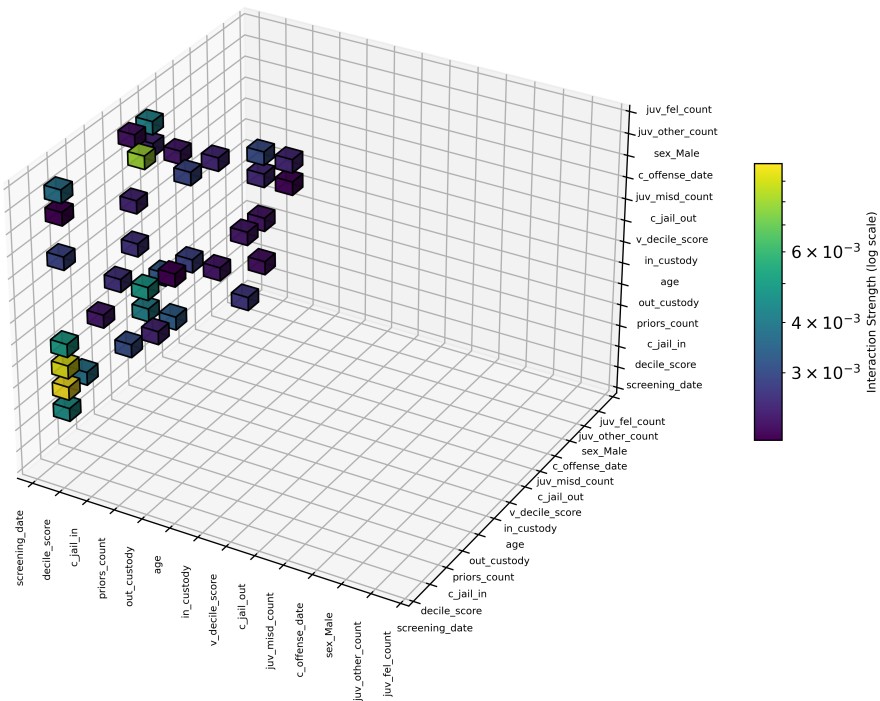

Figure 5: Heatmap of the 36 strongest learned third-order interaction strengths aggregated across field triplets.

## 6  Conclusions

We introduce tensorFM, a novel variant of factorization machines designed to model higher-order cross-interactions between feature embeddings. Our model learns a weighting of these cross-order interactions that is low-rank, enabling efficient computation at inference time. Thereby, tensorFM also provides interpretability by its representation of the cross-interaction strength between different fields. Finally, we empirically demonstrate that our model presents a favorable trade-off between accuracy and inference time compared to existing approaches in the field.

An interesting direction for future work is to integrate our model within a two-stream architecture, combining it with deep neural components as in DCNv2 (Wang et al., 2021) or xDeepFM (Lian et al., 2018). While our focus has been on achieving a favorable trade-off between accuracy and inference time, augmenting our model with deep learning modules may further improve performance, and we leave this extension to future research.

**Broader Impact.** This paper uses the COMPAS dataset for empirical evaluation. Because this dataset encodes sensitive information and reflects structural biases in the U.S. criminal-justice system, any predictive model trained on it risks reinforcing or amplifying these disparities. Our contribution is methodological, and we do not propose deployment in real decision-making settings; moreover, our experiments do not optimize or evaluate fairness metrics. We encourage readers to treat our use of COMPAS as a benchmark study, and to ensure that any downstream use of related models involves domain-expert review, fairness auditing, and alignment with appropriate legal and ethical standards.

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

## A    Higher-Order Singular Value Decomposition

In this section, we discuss the higher-order singular value decomposition of a tensor, and how it can be exploited to obtain a fast computation of the higher-order interactions between fields. In particular, we will prove a similar statement to the one obtained for the CP decomposition (Lemma 1).

Given an $\ell$ order tensor $\mathbf{W} \in \mathbb{R}^{n \times \cdots \times n}$, we say that $\mathbf{W}$ has *multilinear rank* $(r_1, \ldots, r_\ell)$ if there exists a collection of $\ell$ matrices $\boldsymbol{U}^{[1]}, \ldots, \boldsymbol{U}^{[\ell]}$, with $\boldsymbol{U}^{[i]} \in \mathbb{R}^{n \times r_i}$ for $1 \leq i \leq \ell$, and a *core tensor* $\mathbf{S} = \mathbb{R}^{r_1 \times \cdots \times r_\ell}$ such that:

$$\mathbf{W} = \sum_{i_1=1}^{r_1} \ldots \sum_{i_\ell=1}^{r_\ell} \mathbf{S}_{i_1,\ldots,i_\ell} \cdot (\boldsymbol{u}_{i_1}^{[1]} \otimes \ldots \otimes \boldsymbol{u}_{i_\ell}^{[\ell]}) \tag{9}$$

**Lemma 2.** *Let* $\mathbf{W} \in \mathbb{R}^{n \times \cdots \times n}$ *be a $\ell$-order tensor. If* $\mathbf{W}$ *has multilinear rank* $(r_1, \ldots, r_\ell)$, *then it is possible to evaluate*

$$T = \sum_{i_1,\ldots,i_\ell} \mathbf{W}_{i_1,\ldots,i_\ell}^{[\ell]} \cdot \langle \boldsymbol{a}_{\boldsymbol{x},i_1}, \ldots, \boldsymbol{a}_{\boldsymbol{x},i_\ell} \rangle_F$$

*with time complexity* $O((\prod_{i=1}^{\ell} r_i) \cdot k + \ell \cdot nk \max\{r_1, \ldots, r_\ell\})$.

*Proof.* By proceeding as in the proof of Lemma 1, it is possible to show that

$$T = \left\langle \sum_{i=1}^{k} \underbrace{\overline{\boldsymbol{a}}_{\boldsymbol{x},i} \otimes \ldots \otimes \overline{\boldsymbol{a}}_{\boldsymbol{x},i}}_{\ell \text{ times}}, \sum_{i_1=1}^{r_1} \ldots \sum_{i_\ell=1}^{r_\ell} \mathbf{S}_{i_1,\ldots,i_\ell} \cdot (\boldsymbol{u}_{i_1}^{[1]} \otimes \ldots \otimes \boldsymbol{u}_{i_\ell}^{[\ell]}) \right\rangle_F$$

$$= \sum_{i_1=1}^{r_1} \ldots \sum_{i_\ell=1}^{r_\ell} S_{i_1,\ldots,i_\ell} \sum_{h=1}^{k} \prod_{j=1}^{\ell} \left( \boldsymbol{u}_{i_j}^{[j]} \cdot \overline{\boldsymbol{a}}_{\boldsymbol{h}} \right)$$

The computational time to evaluate the above expression is $O((\prod_{i=1}^{\ell} r_i) \cdot k + \ell \cdot nk \max\{r_1, \ldots, r_\ell\})$.  $\square$

## B    Additional Details on the Experiments

In the supplementary material, we provide a `readme` that contains information on how to run our code and reproduce our results. We use a cluster with 48 CPU and 192GB of RAM. It takes one week to run all the experiments with this amount of computational power.

