# OpenReview forum: "tensorFM: Low-Rank Approximations of Cross-Order Feature Interactions"
_TMLR — Rejected by TMLR_

### Review · Reviewer_iC44 · 2025-10-19

**Summary Of Contributions:**

This paper introduces a novel factorization machine variant designed for tabular categorical data prediction tasks. It extends Field-weighted Factorization Machines  by modeling higher-order feature interactions via low-rank Canonical Polyadic  tensor decomposition. Empirically, tensorFM achieves competitive AUC performance on benchmark datasets compared to baselines while maintaining low latency. It also enhances interpretability by aligning learned interaction strengths with mutual information from data.

Strengths

1. Unlike FwFM limited to second-order interactions or deep models, tensorFM uses low-rank tensor decomposition to capture higher-order interactions without exponential complexity.

2. It outperforms or matches baselines on AUC across datasets while scaling linearly with the number of fields during inference.

3. The tensor parameters explicitly encode interaction strengths between fields, and experiments show alignment between learned interactions and data-driven mutual information.

4. It naturally extends to numerical features and supports tasks like binary classification and regression.

Weakness

1. Performance depends heavily on tuning $r$ and $d$; no adaptive mechanism is proposed to automatically optimize these parameters for new datasets.

2. The model avoids deep neural network components to reduce latency, missing potential accuracy gains from combining explicit tensor-based interactions with implicit DNN-learned patterns.

3. While better than naive methods, inference complexity still increases with $d^2$; for $d>4$, latency may become non-trivial for real-time applications.

4. The synthetic dataset for validating higher-order interactions uses only 3–4 fields with limited feature diversity, making it less representative of real-world high-dimensional data.

**Audience:**

Yes

**Audience Explanation:**

This work focuses on core machine learning tasks: predictive modeling for structured data, which falls squarely within the scope of TMLR.

**Broader Impact Concerns:**

The work’s focus on recidivism prediction using the COMPAS dataset and click-through rate (CTR) prediction for online advertising introduces ethical concerns that warrant explicit discussion in a Broader Impact Statement, as these domains carry risks of bias, fairness, and societal harm if models are deployed without safeguards.

**Claims And Evidence:**

Yes

**Claims Explanation:**

The claims made in this paper are supported by accurate, convincing, and clear evidence, primarily through three interconnected pillars: theoretical rigor in complexity derivation, systematic empirical validation across diverse datasets, and quantitative alignment of model behavior with data-driven metrics.

**Requested Changes:**

1. It is recommended to automatically select optimal $r$ and $d$ based on dataset characteristics.

2. Please test tensorFM on datasets with more fields (e.g., n>50) and higher d to quantify latency bounds, and optimize implementation for large-scale deployments.

3. Please generate synthetic datasets with more fields and varying sparsity to better validate the model’s ability to capture higher-order interactions in realistic scenarios.

---

> ### Author Response · Authors · 2025-11-18
> **Response**
>
> We thank the reviewer for their valuable feedback and for their time spent reviewing our paper. We updated our paper according to the feedback provided, and highlighted with blue text color the modified parts for ease.
>
> ---
>
> **Reviewer**: It is recommended to automatically select optimal
>  $r$ and  $d$ based on dataset characteristics. Performance depends heavily on tuning of $r$ and $d$; no adaptive mechanism is proposed to automatically optimize these parameters for new datasets.
>
>  **Answer**: One can treat $r$ and $d$ as hyper-parameters, and use a validation set to find the optimal choice. Of course, this choice of $r$ and $d$ will affect the inference complexity as well.  In the results of Table 2, we actually use this hyper-parameter search, and the reported low-complexity TensorFM and high-complexity tensorFM use different values of $r$ and $d$ across the two different datasets Avazu and Criteo, as described in Section 5.1
>
> Additionally, in the plots reported in Figure 1 and Figure 2, where we show the difference in model performance varying $r$ and $d$.
>
> ---
>
> **Reviewer**: Please test tensorFM on datasets with more fields (e.g., n>50) and higher d to quantify latency bounds, and optimize implementation for large-scale deployments.
>
> **Answer**: In Table 9, we generate a synthetic dataset with a large number of fields $(n=100)$. For measuring the inference complexity in that table, we use an implementation of the methods that is more efficient than querying naively the pytorch model.  Unfortunately, we were not able to find publicly available datasets with substantially larger numbers of fields.
>
> ---
>
> **Reviewer**: Please generate synthetic datasets with more fields and varying sparsity to better validate the model’s ability to capture higher-order interactions in realistic scenarios.
>
> **Answer**: We generated a synthetic dataset with a larger number of fields $(n=100)$ and updated the section 5.3 accordingly.
> We tested all baseline models on this dataset and reported their Log-Loss and AUC scores in the section. These results show that tensorFM can effectively learn high-order interactions even when there are many "noisy" (random) columns, demonstrating its robustness.
>
> ---
>
> **Reviewer**: The model avoids deep neural network components to reduce latency, missing potential accuracy gains from combining explicit tensor-based interactions with implicit DNN-learned patterns.
>
> **Answer**: We thank the reviewer for this feedback. As emphasized in the paper (at the end of the Related Work section), our goal is not to propose the most accurate model but one that achieves a favorable trade-off between accuracy and inference time. Incorporating deep learning components similar to state-of-the-art models such as DCN_V2 could further enhance performance, and we view this as a promising future research direction beyond the current scope of our work.

---

> > ### Comment · Reviewer_iC44 · 2025-12-16
> > **Thank you for the response.**
> >
> > Thank you for the response. You have addressed the concerns raised, and I have no further questions.

---

### Review · Reviewer_NZZP · 2025-10-21

**Summary Of Contributions:**

This paper focuses on modeling prediction problems on tabular datasets with categorical features. Simple methods may only consider single-order interactions (e.g., regression) which limits the modeling capacity. Therefore, other methods, such as factorization machines (FMs), attempt to compute higher-order interactions between features. This is done by associating a learnable embedding with each feature, the interactions of two features is thus given by the dot product. Furthermore, learnable weights can be used to weight the strength of those interactions. There is, however, a trade-off here, as the number of learnable weights scales with the number of interactions. This means that if we do consider higher-order interactions (e.g., beyond 2nd order), we will have many weights to learn. More recently, neural network approaches have been used to model such interactions, however the authors argue that they are inherently inefficient and don't scale. As such, the authors seek an approach to efficiently consider higher-order feature interactions without the use of deep neural networks.

To this point, the authors introduce tensorFM. Their main contribution is that the weight matrix $S^{[l]}$, the models the interactions for some order $l$, can be constrained to have some rank of $r$ via CP factorization. This is noteworthy as it can reduce the complexity from $O(n^2 k)$ to $O(rnk)$ where $r$ is typically much smaller than $n$ (I omit $d$ here for simplicity). This thus allows the authors to scale their FM to a max order of up to $d=4$ in their experiments. The authors show results on multiple datasets, showing some promising results. They also include efficiency studies to motivate the scalability of their method while including a case study that highlights the interpretability of the model.

### Strengths:

1. The paper is very clearly written. It does a very good job of explaining the overall problem, existing work, and how their method can solve this problem while being distinct from earlier methods.

2. The method, as its core, is quite simple. This is a good thing, as in my opinion, simpler methods are always preferred to those that are more complicated. It makes implementing and training the method to be much easier and practice. Therefore, this is a definite strength for me.

3. The authors test on a variety of real-world datasets, including experiments on synthetic data. This gives us a good idea of the performance of the method compared to baselines.

4. I further appreciate that authors include multiple ways of measuring the efficiency in the FLOPs, time, and complexity. This allows for a multifaceted view on the scalability of their method.


### Weaknesses:

1. The performance improvement is quite marginal. To be clear, I don't think that SOTA performance is necessary for a good method. But from a performance perspective, it seems that there is very little reason to use TensorFM. This also calls into question how often higher-order interactions ($d > 2$) are needed, as the returns seem very diminishing.

2. As a corollary to the previous point, the variance across model runs is not reported. This is despite averaging performance over 3 runs on Avazu and Criteo. The problem here is that the difference in performance versus AFM, FwFM, and CN is quite small. Therefore, to determine if the performance increase is actually significant, the variance must be reported, with the statistical significance calculated.

3. Limited discussion of (Shtoff et al., 2024b). The authors mention that TensorFM is a "direct generalization of the idea to higher orders of field interaction tensors" (page 4). However, a limited comparison is given. Given that the authors state that they indeed are highly connected, it makes sense that more time is given to how they are and how TensorFM generalizes it.

4. The efficiency comparison shown in Table 6 is not terribly convincing. While TensorFM is *generally* faster than FwFM and CN, all the methods already run extremely fast. As such, is efficiency really that big of a problem here? Furthermore, while the authors use $n=100$ in their experiments, the datasets have $n$ of 22, 39, and 14. So this is clearly larger. I'd recommend showing the time on those datasets, as they are closer to what we would see in the real-world. To be clear, any decrease in time is good (all things considered), but at the end of the day, most models already seem fairly fast.

5. This is a minor weakness, but it's unclear to me what we should take away from the experiments using synthetic data. As shown, for 3rd order interactions AFM does better than TensorFM. The opposite is true for 4th order. Why? What are the authors trying to claim here? I recommend revising this part to be clearer.

**Audience:**

Yes

**Audience Explanation:**

The paper addresses an important area of research in tabular prediction. More generally, FMs are very commonly used in recommender systems where many recommendations need to be done in real time. Therefore, improving efficiency is highly important. As such, I'd argue this is a relevant topic to the ML community.

**Claims And Evidence:**

Yes

**Claims Explanation:**

The main claims in the paper are backed up with solid evidence. Essentially: (1) The authors show that their performance can achieve comparable or slightly better performance than baselines ; (2) The method is indeed more efficient than other FMs ; (3) There is an indication that it can model data with higher-order interactions better than baselines. That said, there are also a few weaknesses in the evidence. I expound on them in the requested changes section below.

**Requested Changes:**

1. Please include the variance over runs in Tables 2 and 4. This allows us to determine if the performance increase is statistically significant.
2. Please expound on the findings from the synthetic data experiments. Specifically, detailing what they reveal to us and why they are important. As noted in weakness 5, it's currently unclear what exactly they tell us.
3. Please provide more details on the relationship of the proposed method to (Shtoff et al., 2024b). the authors themselves note that their method is a higher-order generalization of it. As such, more time should be devoted to their relationship and how TensorFM generalizes their method.
4. The time efficiency is taken over synthetic data with $n$ that is larger than each of the real datasets. What about the efficiency on the three datasets use? I'd argue this is a much better proxy for real-world expectations, as I'm unsure how common $n=100$ even is. Furthermore, it could tell us the expected speed-up for different values of $n$, which can be valuable in practice.

---

> ### Author Response · Authors · 2025-11-18
> **Response**
>
> We thank the reviewer for their valuable feedback and for their time spent reviewing our paper. We updated our paper according to the feedback provided, and highlighted with blue text color the modified parts for ease.
>
> ---
>
> **Reviewer**: More details on the relationship of the proposed method to (Shtoff et al., 2024b).
>
> **Answer**: We thank the reviewer for this feedback. We updated the draft of our paper with a longer discussion on the method proposed by Shtoff et al., 2024b in the related work section. We highlight out that two discussed previous work (Almagor & Hoshen, 2022; Shtoff et al., 2024b) use similar low rank ideas to achieve low computational complexity at inference time, but their approach is limited to $d=2$, and our work extends those ideas to higher-order interactions.
>
> Revised paragraph:
>
> A similar low-rank decomposition of the field-interaction matrix has also been proposed in other recent work \cite{shtoff2024low}, where the authors introduce the interaction matrix
> $S = U \Lambda U^T[ (U \Lambda U^\top) \odot I_n\bigr],$
> with $U \in \mathbb{R}^{n \times r}$, $\Lambda \in \mathbb{R}^{r \times r}$ is diagonal, and $\odot$ denotes the element-wise (Hadamard) product.
> This construction is closely related to our model, with the additional modeling constraint that $S$ has zero-diagonal}. Our model generalizes the ideas of the aforementioned papers beyond second-order interactions ($d=2$) and can thus capture higher-order interactions.
>
>
> ---
>
> **Reviewer**: Include variance over runs.
>
> **Answer**:   We have to apologize for the incorrect description of Table 2 stating that the average test set AUC (%) was reported based on 3 runs (this sentence has not been updated from our previous submission, for which we used a different codebase and hyperparameter tuning strategy).
> We indeed performed an extensive variance analysis early during our investigations that was convincingly showing very low standard deviations between separate training runs. Therefore, for the final experiments reported in Table 2, we decided to make the most out of the training runs by increasing the hyperparameter range instead of running repetitions.
>
> ---
>
> **Reviewer**: Expand on the synthetic data experiments and revisit the section.
>
> **Answer**:
> We thank the reviewer for this feedback.
> We updated the section with a newer result for $n=100$ columns.
> Our syntetic data is generated by random values in the columns where only 3 (4) columns uniquely define the labels.
> In this way we compare different models being able to learn high order feature interactions among completely random values and many noisy columns.
>
> ---
>
>
> **Reviewer**: The efficiency comparison shown in Table 6 is not terribly convincing. While TensorFM is generally faster than FwFM and CN, all the methods already run extremely fast. As such, is efficiency really that big of a problem here? Furthermore, while the authors use n=100 in their experiments, the datasets have of 22, 39, and 14. So this is clearly larger. I'd recommend showing the time on those datasets, as they are closer to what we would see in the real-world. To be clear, any decrease in time is good (all things considered), but at the end of the day, most models already seem fairly fast.
>
> **Answer**: We agree that all models run fast at the scale of current public datasets. Unfortunately, we were not able to find publicly available datasets with substantially larger numbers of fields. However, in real production systems the number of fields can grow significantly, and expanding the feature space is a core part of feature engineering pipelines. This is our motivation for reporting inference complexity with $n=100$.
> Our model is designed to scale linearly in the number of fields by using a low-rank tensor decomposition, whereas several baselines incur quadratic or worse interaction costs. This linear scaling is the main reason TensorFM remains fast when $n$ grows (as shown in the FLOPs plot in Figure 3), and why we believe efficiency becomes a practical concern in larger real-world settings.

---

### Review · Reviewer_fvbK · 2025-11-06

**Summary Of Contributions:**

This paper introduces tensorFM, a variant of Factorization Machines that models cross-field interactions up to order $d$ by learning low CP-rank tensors $S^{[\ell]}$ representing the interaction strengths. The primary technical contribution is a theoretical proof that, under a CP decomposition of each interaction tensor, the model's inference time scales as $O(n k \sum_{\ell=2}^{d} \ell r_{\ell})$ (Theorem 1), enabling explicit high-order interactions with low latency. Empirically, tensorFM is evaluated on real-world datasets (Avazu, Criteo, COMPAS) and synthetic data designed with pure 3- and 4-way interactions, showing competitive AUC and favorable FLOPs/runtime compared to several lightweight baselines.

$\textbf{Key Strengths:}$
1. Conceptual clarity, presenting a natural and clear generalization of Field-weighted FM (FwFM) to higher-order interactions.

2. Sound derivations (e.g., Lemma 1) and an interpretable complexity bound.

3. Reasonable empirical scope, covering both click-through-rate prediction and interpretability-sensitive scenarios.

4. The efficient inference claim is well-supported by both FLOPs analysis and empirical runtime measurements.

$\textbf{Key Weaknesses:}$

1. Omission of the most relevant high-order baseline (HOFM/Polynomial Networks).

2. Small performance gains on real-world datasets are reported without statistical significance testing.

3. Ambiguity in the definition of the interaction tensors $S^{[l]}$ (e.g., symmetry, handling of self-interactions), which impacts interpretability and reproducibility.

**Additional Comments:**

The core idea of the paper—learning higher-order interactions via low-rank tensor decompositions—is clear and promising. The theoretical derivation and efficiency analysis are significant strengths. The primary barriers in the current version are the completeness and rigor of the empirical evaluation. Should the authors address the critical changes outlined above, particularly the inclusion of HOFM baselines and statistical testing, the persuasiveness of the paper's contributions would be greatly enhanced. I look forward to seeing a revised version.

**Audience:**

Yes

**Audience Explanation:**

Yes, a significant portion of TMLR's audience would be interested in these findings. Research into efficient and interpretable models for tabular data, particularly for core applications like click-through-rate prediction and recommender systems, is a central and active area within the machine learning community. This paper presents a direct generalization of the Factorization Machine family and provides a theoretical complexity analysis alongside promising empirical results. Its focus on low latency and higher-order interactions holds clear appeal for researchers and practitioners working on large-scale ML systems, recommender systems, and those requiring model interpretability.

**Broader Impact Concerns:**

The paper utilizes the COMPAS dataset, which involves sensitive societal attributes related to criminal justice and race. While the paper is primarily methodological, any predictive model built on this dataset can have real-world implications for individuals.

Recommendation: The paper should add a "Broader Impact" statement or paragraph. This should briefly discuss the potential fairness risks of applying such models in sensitive domains like COMPAS (e.g., the potential for models to amplify historical biases present in the data). Even if fairness metrics were not directly optimized in the experiments, acknowledging this limitation and encouraging responsible deployment is necessary.

**Claims And Evidence:**

Yes

**Claims Explanation:**

The paper's core claims are supported by compelling evidence:

1.  The theoretical complexity of $ O(nk\sum_{\ell=2}^{d}\ell r_{\ell}) $ (Theorem 1) is convincingly validated by the FLOPs analysis (Figure 3) and runtime measurements (Table 6), which show linear scaling and superior speed over quadratic baselines like FwFM.

2. The synthetic data experiments (Table 5) provide clear and direct evidence that the model effectively captures genuine 3- and 4-way interactions, significantly outperforming methods limited to lower-order interactions.

3. The methodology in Section 5.6 offers a concrete and well-executed analysis, demonstrating a quantitative correlation with mutual information and providing qualitative insights (Figure 5).

The primary limitation concerns the breadth of the "competitive performance'' claim, which requires comparison to HOFM baselines and statistical significance tests for full validation. However, the evidence for the central technical contributions regarding efficiency, capability, and interpretability is accurate, clear, and convincing on its own.

**Requested Changes:**

$\textbf{Critical Changes (Essential for Acceptance):}$

1.  It is critical to include Higher-Order Factorization Machines (HOFM) or polynomial/ANOVA-kernel FMs as baselines, comparing their performance and inference time against tensorFM on the Avazu and Criteo datasets.

2.  For the results in Table 2 and Table 4, report the mean and standard deviation over multiple runs and perform statistical significance tests (e.g., paired t-tests) to substantiate the claimed performance improvements.

3.  The text must explicitly state:

$\quad$ 3.1 In Equation (7), whether the indices $i_1, \dots, i_\ell$ are required to be distinct (i.e., excluding self-interactions).

$\quad$  3.2 Whether the tensor $S^{[\ell]}$ is constrained to be super-symmetric. If not, the rationale and how permutation sensitivity is addressed should be explained.

$\quad$ 3.3 For the $\ell=2$ case, whether $S^{[2]}$ is constrained to be symmetric with a zero diagonal (for a fair comparison with FwFM). The method of enforcing these constraints in the parameterization and code (e.g., through masking or tying) must be described.

$\textbf{Recommended Changes (Would Strengthen the Work):}$

4.  Specify the optimizer, loss function, learning rate schedule, initialization method for CP factors, and use of early stopping. Additionally, discuss and compare the training time and memory consumption relative to baselines like FwFM.

5. Fix Notation and Typos:

$\quad$5.1 Page 3: duplicated clause “one feature for each field.active features…”.

$\quad$5.2 Page 5: “equation equation 5”.

$\quad$5.3 Spelling: “matricies” → “matrices”, “featuers” → “features”.

$\quad$5.4 Notation consistency: Equation (8) uses 𝒲 vs bold W; keep consistent.


6. Enhance Experimental Analysis:

$\quad$6.1 Consider adding LogLoss results for Avazu and Criteo to enable comparison with a broader literature.

$\quad$6.2 The claim of $\Omega(n^3)$ complexity for CIN in Table 6 should be justified with a derivation or standard reference, or revised to a more widely accepted complexity expression.

$\quad$6.3 Include ablations on the embedding dimension $k$ and the trade-off between different $(d, r)$ combinations on at least one real-world dataset.

---

> ### Author Response · Authors · 2025-11-18
> **Response**
>
> We thank the reviewer for their valuable feedback and for their time spent reviewing our paper. We updated our paper according to the feedback provided, and highlighted with blue text color the modified parts for ease.
>
>
> ---
>
> **Reviewer**: Inclusion of HOFM
>
> **Answer**: We are working on adding experiments with the HOFM baseline to our paper. On the first revision, we already added results for synthetic data and runtime.
>
>
> ---
>
> **Reviewer**: Reporting standard deviation for experiments.
>
> **Answer**: We have to apologize for the incorrect description of Table 2 stating that the average test set AUC (%) was reported based on 3 runs (this sentence has not been updated from our previous submission, for which we used a different codebase and hyperparameter tuning strategy).
> We indeed performed an extensive variance analysis early during our investigations that was convincingly showing very low standard deviations between separate training runs. Therefore, for the final experiments reported in Table 2, we decided to make the most out of the training runs by increasing the hyperparameter range instead of running repetitions.
>
> ---
>
> **Reviewer**: About the super-symmetry of the tensors and self interactions for the tensor $S^{[l]}$.
>
> **Answer**: We updated the paper to include a discussion about the symmetry of the tensor. We do not add any symmetry assumption in our modeling. However, note that for each non super-symmetric tensor, there exists a symmetric tensor that yields the same output in our model. Thus, there is no loss of expressivity moving from symmetric tensor to non symmetric tensor and  viceversa.
>
> We also do not constraint the diagonal to be zero in the case $d=2$. This contrasts a difference with the FwFM model. While it would be possible to remove the diagonal in the case $d=2$, we did not find any easy extension to remove the self-interactions for d > 2 while still preserving the computational efficiency.
>
> ---
>
> **Reviewer**: Different combination of (r,d) on real datasets and ablation for choice of $k$.
>
> **Answer**: We provide  plots varying (r,d) for Avazu in Section 5.4. We decided to fix the value $k$ across all experiments since this is a shared hyper-parameter among the baseline models.
>
> ---
>
> **Reviewer**: Information on hyper-parameters, and optimizer.
>
> **Answer**: We use optuna for the hyper-parameter search, and train for a fixed number of epochs as we detail in Section 5.  We added details on the optimizer in that section, we thank the reviewer for their feedback.
>
> ---
>
> **Reviewer**: Consider adding LogLoss results for Avazu and Criteo to enable comparison with a broader literature.
>
> **Answer**: We will try to re-run the experiments to obtain the log loss results.
>
> ---
>
> **Reviewer**:  The claim of complexity for CIN in Table 6 should be justified with a derivation or standard reference, or revised to a more widely accepted complexity expression.
>
> **Answer**: We provide additional explanation on the related work section. The result is taken from the CIN paper. From section 3.1 of the CIN paper (Lian et al, 2018), where they describe the polynomial approximation capability of the model, they use the hyperparameter H=n (where n is the number of fields). The time complexity of CIN is $\Omega(n H^2) = \Omega(n^3)$.
>
>
> ---
>
> **Reviewer**: Broader impact statement due to using COMPAS dataset.
>
> **Answer**: We thank the reviewer for this feedback. We added a statement in the conclusion.
>
> Added paragraph:
>
> \textbf{Broader Impact.}  This paper uses the COMPAS dataset for empirical evaluation. Because this dataset encodes sensitive information and reflects structural biases in the U.S. criminal-justice system, any predictive model trained on it risks reinforcing or amplifying these disparities. Our contribution is methodological, and we do not propose deployment in real decision-making settings; moreover, our experiments do not optimize or evaluate fairness metrics. We encourage readers to treat our use of COMPAS as a benchmark study, and to ensure that any downstream use of related models involves domain-expert review, fairness auditing, and alignment with appropriate legal and ethical standards.
>
> ---
>
> We also thank the reviewers for pointing out the typos, that have been fixed in the updated version.

---

### Review · Reviewer_vR65 · 2025-11-27

**Summary Of Contributions:**

The paper introduces TensorFM, a new model for capturing higher-order interactions between categorical attributes. The model is based on the CP decomposition of a higher-order tensor. The paper can also be seen as a higher-order variant of FwFM, which is a low-rank approximation of second-order interactions. The novelty comes from CP decomposition as they can reduce the complexity of the higher order interactions from $O(d^2kn^d)$ to $O(nk\sum_{l=2}^dlr_l)$.

**Audience:**

Yes

**Audience Explanation:**

This work addresses the prediction problem and is one of the methods in factorization machines, which have applications in recommender systems and online advertising, both subareas of machine learning. Therefore, it would be interesting for the TMLR's audience.

**Broader Impact Concerns:**

Thanks for adding the broader impact. Agreed with other reviewers.

**Claims And Evidence:**

Yes

**Claims Explanation:**

Yes, the claims and contributions seem convincing and well clarified. However, the definition of the tensor $\mathbf{S}^{[(l)]}$ is not well presented.
I didn't understand why you haven't compared HOFM as a baseline with your method, while you've written in blue and added a comparison with it for only the ablation study. Also, do you have any studies showing whether there is any trade-off between rank and performance? Curious about Table 7 and Figure 3. What is the role of increasing the rank? As it increases the complexity. There are also some ambiguities that I will write in the "Requested Changes" section.

**Requested Changes:**

- Introduction:
1. FFM stands for Field-aware Factorization Machines? If yes, write it in front of it.
- Preliminaries:
1. Kronecker product is not defined.
2. A tensor is defined as a p-th order tensor with $n_1\times\cdots\times n_p$, which is not consistent in notation with the tensor defined in section 3.2.
3. I couldn't get why we should have a sigmoid function definition while we don't use it throughout the text.
4. Line 5, X should be bold.
5. Highly dimensional -> high-dimensional
- TensorFM algorithm:
1. In 3.1, isn't $\mathbf{S}$ also symmetric zero-diagonal parameter matrix for FM as well?
2. In proposition 1, specify the dimension of $\mathbf{A}_{\mathbf{x}}$.
3. $A_xU$, $A_xV$, $I_n$ and $1_n$ should be in bold.
4. In 3.2, the definition of tensor $\mathbf{S}^{[(l)]}$ is very confusing.

---

> ### Author Response · Authors · 2025-12-05
> **Response**
>
> We thank the reviewer for their time spent reading our paper and for their feedback.
>
> ---
>
> **R**: The definition of the tensor $S^l$ is not well presented.
>
> **A**: We slightly changed the wording around the definition. For tensorFM, the definition of the model is formally given in Section 3.3 after introducing the required notation such as CP rank. In particular, for any $2 \le l \le d$, we have parameters $U^l_1$ ... $ U^l_l$  which are real matrices of dimension $n \times r_l$. Given these parameters, the tensor $S^{l}$ is defined as $S^l$ $=\sum_{i=1}^{r_l}$ $u^l_{1,i}$ $\otimes$ ... $\otimes$  $u^{l}_{l,i}$.
>
> ---
>
> **R**:  Also, do you have any studies showing whether there is any trade-off between rank and performance? Curious about Table 7 and Figure 3. What is the role of increasing the rank? As it increases the complexity.
>
> **A**: Larger value of $r$ and $d$ yield a more complex model, which can possibly capture additional signal from the data. There is a trade-off: higher-complexity can possibly obtain better performance (as shown in Figure 1 and 2), but it has an higher computational cost (Figure 3, Table 7).
>
> ---
>
> We addressed the requested changes, thank you for the feedback. Some short replies are provided below.
>
> ---
>
> **R**: In 3.1, isn't S also symmetric-zero diagonal parameter matrix for FM as well?
>
> **A**: Yes, it is symmeteric zero-diagonal for both FwFM and FM. For FM, this is a fixed matrix, while for FwFM it is a parameter. We changed the wording to make this clearer, thanks.
>
> ---
>
> **R**: I couldn't get why we should have a sigmoid function definition while we don't use it throughout the text.
>
> **A**: We only specified that we obtain the prediction through the sigmoid function. The definition is just to make the notation easier, but we agree it is not necessary.

---

### Author Response · Authors · 2025-11-18
**General Comment**

Thank you everyone for your time spent reviewing our paper and for your feedback.

Below we give responses to individual reviewers. Some of our answers are still pending, particularly those that require additional experiments. We will follow up on those in coming days.

---

### Author Response · Authors · 2025-12-16
**Manuscript Update**

We thank the reviewers for their time and constructive feedback. We have revised the manuscript accordingly and added new experiments including the additional baseline HOFM.  For ease, we report the table with the updated results on Criteo and Avazu. Other major modifications are highlighted in blue in the updated version.

| Model | Avazu Test Log-Loss | Avazu Test AUC (%) | Criteo Test Log-Loss | Criteo Test AUC |
|---|---:|---:|---:|---:|
| LR | 0.3873 | 76.53 | 0.4551 | 79.39 |
| FM | 0.3810 | 77.69 | 0.4470 | 80.44 |
| FwFM | 0.3805 | 77.28 | 0.4428† | 80.87† |
| AFM | 0.3833 | 77.31 | 0.4462 | 79.75 |
| CN | 0.3841 | 77.16 | 0.4514 | 79.55 |
| HOFM | **0.3797** | **77.91** | 0.4449 | 80.67 |
| low-complexity tensorFM | 0.3805 | 77.74 | 0.4436 | 80.79 |
| high-complexity tensorFM | 0.3804† | 77.77† | **0.4422** | **80.94** |

---

### Decision · Action_Editor_PAWU · 2026-01-14

**Recommendation:** Reject

**Audience:**

Yes

**Audience Explanation:**

Regardless of the final recommendation for the paper, modeling structured/tabular data efficiently continues to remain an important topic in machine learning, particularly for recommender systems. The paper does present a clean and theoretically grounded generalization of factorization machines to higher-order interactions, with explicit attention to inference efficiency and interpretability. Several reviewers agree that these contributions are relevant and potentially useful to both researchers and practitioners within TMLR’s audience. However, the weaknesses pointed out by the reviewers also make it hard to justify the paper's inclusion.

**Claims And Evidence:**

No

**Claims Explanation:**

The core technical claims, i.e., tensorFM efficiently models higher-order feature interactions via low-rank tensor decompositions with favorable inference complexity, have reasonable support by theoretical analysis and empirical results. Multiple reviewers agree that the complexity bounds, efficiency analysis, and synthetic experiments convincingly demonstrate the model's ability to capture higher-order interactions.

However, there are also concerns remain regarding the empirical rigor of real-world performance comparisons. More specifically, several reviewers note the absence of variance reporting over multiple runs and question whether the relatively small performance gains are statistically significant. While the authors provided clarifications and added baselines (including HOFM) and additional experiments, at least two reviewers remain unconvinced that the empirical evidence fully substantiates the strength of the performance claims. In particular, the multiple reviewers remained critical because the requested variance results/multiple seed experiments were not provided and only rather vague explanation/justification was provided.

Although some reviewers view the contribution favorably and acknowledge its clarity and potential interest, the lingering concerns by at least two reviewers prevent a confident assessment of the claimed empirical improvements. As a result, the submission does not meet the acceptance bar.

Given the positive aspects, however, if the authors wish do, they could consider resubmitting as a major revision in future, provided the issued raised by the reviewers of the current submission are properly addressed.

**Resubmission Of Major Revision:**

The authors may consider submitting a major revision at a later time.